# Research on Seismic Performance and Reinforcement Methods for Self-Centering Rocking Steel Bridge Piers

Hanqing Zhuge [1,*], Chenpeng Niu [1], Rui Du [1] and Zhanzhan Tang [2]

1 College of Civil Engineering and Architecture, Zhejiang University of Science & Technology, Hangzhou 310023, China; 212202814031@zust.edu.cn (C.N.); durui188hz@gmail.com (R.D.)
2 College of Civil Science and Engineering, Yangzhou University, Yangzhou 225009, China; zztang@yzu.edu.cn
* Correspondence: 120043@zust.edu.cn

**Abstract:** To study the seismic performance of self-centering circular-section rocking steel bridge piers whose functions can be restored after an earthquake, a high-precision finite element (FE) analysis model of such a bridge piers was established. The hysteresis behavior of concrete-infilled and hollow rocking steel bridge piers was compared. In response to the characteristics of the local deformation of the wall plates and elliptical deformation of the bottom surface, two reinforcement methods for the pier bottom, namely thickening the wall plate and adding longitudinal stiffeners in the plastic zone of the pier bottom, were proposed. The pseudo static analysis of bridge piers was carried out considering the effects of overall design parameters and reinforcement parameters of the pier bottom. The results indicate that the FE model used in this paper can obtain accurate horizontal load-displacement curves of rocking steel bridge piers. The hysteresis curves of the rocking steel bridge piers and infilled concrete rocking steel bridge piers is close, and directly using hollow steel bridge piers can improve the economic efficiency of the design. Compared to adding longitudinal stiffeners, the reinforcement form of thickened wall plates at the pier bottom has a better effect in improving the seismic performance of bridge piers. The reinforcement of the pier bottom has little effect on the energy dissipation capacity of the bridge pier, but it helps to reduce residual displacement and improve lateral stiffness.

**Keywords:** bridge engineering; self-centering; rocking steel bridge piers; seismic performance; reinforcement methods

## 1. Introduction

The bridge piers serve as lateral resistance and energy dissipation components in girder bridges [1]. Traditional ductile seismic design concepts may lead to significant residual displacements in bridge piers after an earthquake, causing difficulties in post-earthquake repairs [2,3]. Mander et al. [4] were the first to apply unbonded-prestressing technology to rocking bridge piers, proposing the design concept of prestressed self-centering rocking bridge piers. Subsequently, Pampanin et al. [5] and Palermo [6] proposed the monolithic beam analogy and the modified monolithic beam analogy, enabling the theoretical calculation of the load-displacement envelope curve for rocking bridge piers. To enhance the energy dissipation capacity of bridge piers, Palermo et al. [7] proposed the addition of embedded energy-dissipating steel reinforcement in rocking bridge piers, which significantly enhanced the load-bearing capacity and energy dissipation capability of the piers. However, it also increases residual displacement, and at the same time, replacing the internal energy-absorbing steel bars becomes challenging after an earthquake. Marriott et al. [8] proposed the use of externally replaceable steel rods as a means of energy dissipation for bridge piers. Subsequently, other scholars have also put forward externally placed energy-dissipating steel bars [9], externally placed energy-dissipating aluminum bars [10], and shape memory alloy of energy-absorbing bars [11], all of which greatly improved the post-earthquake

repair ability performance of prestressed rocking bridge piers. Han et al. [12] found through experimental studies that externally installed buckling-resistant energy-dissipating steel plates significantly enhance the energy dissipation capability of bridge piers compared to externally installed energy-dissipating steel rods.

However, the pier bodies of existing reinforced concrete rocking bridge piers have relatively weak energy dissipation and an anti-toppling capacity. Some scholars have conducted research on concrete-filled rocking bridge piers with better ductility performance. For example, Liu et al. [13] conducted pseudo-static comparative tests on prestressed rocking concrete-filled steel bridge piers, prestressed rocking reinforced concrete piers, and socket prestressed concrete-filled steel bridge piers. It was found that the prestressed rocking concrete-filled steel bridge pier exhibited the best seismic performance. Wang et al. [14] performed numerical simulation analysis on prestressed concrete-filled steel bridge piers under three different boundary conditions, which are hinged connection, semi-rigid connection and rigid connection. In addition, other scholars have conducted seismic performance tests on segmentally assembled prestressed rocking concrete-filled steel bridge piers [15,16].

Compared to reinforced concrete and concrete-filled steel bridge piers, hollow steel structure piers can not only avoid local crushing of concrete, but also have advantages such as lightweight, high strength, and environmental sustainability. In recent years, Chen et al. [17], Zhuge et al. [18–20], Li et al. [21], and Li et al. [22] proposed ductile design methods for steel bridge piers, filling the gap in seismic research on steel bridge piers in China. Ahmad et al. [23–25] studied the seismic performance of self-centering rocking steel bridge piers by conducting pseudo-static tests and finite element (FE) calculations for the first time. They found that properly designed piers exhibit excellent ductility and resilience. However, elliptical deformation at the bottom surface and local buckling deformation at the plastic damaged zone may occur, which affect the seismic performance and post-earthquake repair ability of the piers. Currently, this new type of pier structures is both seismic resilient and cost-effective, which makes it promising for application in high-intensity seismic areas. However, further research is needed to explore the seismic performance and design calculation methods.

This study focuses on circular-section prestressed self-centering rocking steel bridge piers. Two bottom reinforcement methods were proposed, which are thickening the steel plate at the bottom and adding longitudinal stiffeners. Numerical simulations were conducted to analyze the influence of overall design parameters and bottom reinforcement parameters on the bridge pier. The research findings provide important insights for the development, design and application of self-centering rocking steel bridge piers.

## 2. Self-Centering Rocking Steel Bridge Pier Structure and Its Finite Element Analysis Model

### 2.1. The Form and Design Parameters of the Structure

Currently, mature self-centering rocking bridge pier structures achieve the rocking motion by releasing the bottom constraint with bases, thus extending the natural period of the structure and providing the vibration isolation effect. Meanwhile, under the axial compression from prestressed tendons and the gravity loads, the bridge piers can be self-centering. The structure undergoes cyclic energy dissipation through the plastic deformation of replaceable energy dissipating components.

Based on the existing research, this paper designs a single-column self-centering rocking steel bridge pier with a circular cross-section, as shown in Figure 1. The pier material adopts the commonly used Q345qC steel in bridge engineering. To achieve better self-centering and energy dissipation functions, prestressed tendons and energy dissipating components are placed at the center of the section and outside the base plate, respectively. The prestressed tendons consist of several strands of $1 \times 7$ prestressed steel wires (each strand having a nominal diameter of 15.2 mm) with a yield strength of 1860 MPa. The initial tension is set to 0.4 times of the yield strength. The energy dissipating component

is an externally attached buckling-restrained energy dissipating steel plate made of Q235 steel with a low yield point.

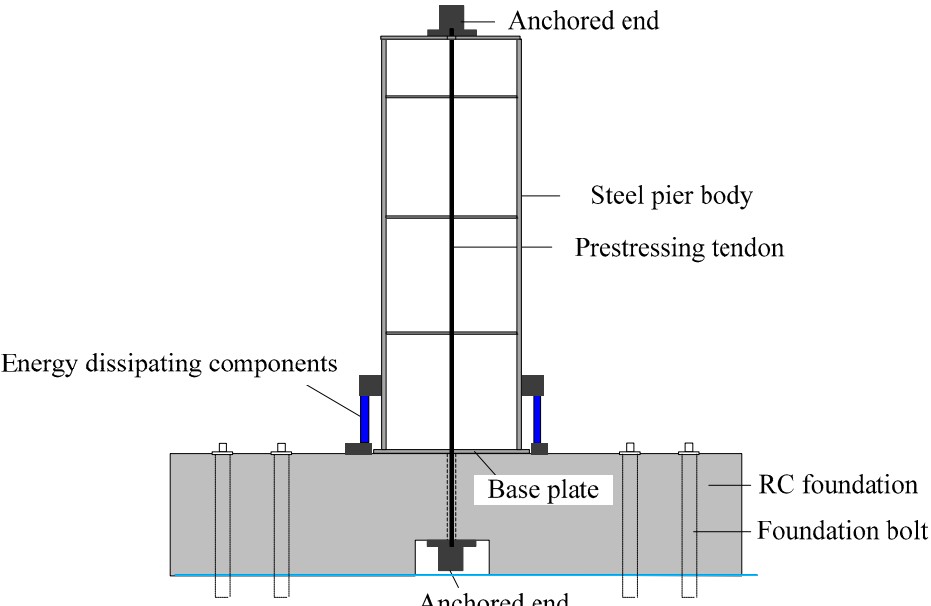

**Figure 1.** Circular-section rocking steel bridge pier structure.

According to the research on seismic performance of circular-section steel bridge piers [17,20], the structural parameters that affect the ductility performance of circular-section steel bridge piers are overall slenderness ratio $\bar{\lambda}$, radius-to-thickness ratio $R_t$ and axial compression ratio $N/N_y$. The formulas for calculating the overall slenderness ratio $\bar{\lambda}$ and radius-to-thickness ratio $R_t$ are as follows:

$$\bar{\lambda} = \frac{2h}{R} \cdot \frac{1}{\pi} \sqrt{\frac{\sigma_y}{E}} \tag{1}$$

$$R_t = \frac{R}{t} \cdot \frac{\sigma_y}{E} \sqrt{3(1-\mu^2)} \tag{2}$$

In the above equations, $h$ represents the pier height, $R$ represents the radius of the cross section corresponding to the center line of the wall plate, and $t$ represents the thickness of the wall plate. $\sigma_y$ and $E$ represent the yield strength and elastic modulus of the steel material, respectively.

According to the seismic performance experimental study conducted by Ahmad et al. [24] on rocking steel bridge piers, the addition of the base plate at the bottom of the rocking steel bridge pier significantly enhances its seismic performance. The enhancement mechanism can be summarized in several aspects. Firstly, the base plate applies tension to the wall plate, counteracting part of the compression force and reducing local deformation effects. Further, the area under shear force at the bottom of the pier increases, making it less prone to elliptical deformation and reducing the effect of stiffness reduction. At the same time, the height of the compression zone decreases. It increases the lever arm for resisting horizontal force and allows the pier to be raised higher. The self-centering effect under the action of prestressed reinforcement is enhanced. However, the influence of base plate thickness and area on the seismic performance is not yet clear. To this end, in this paper, the increasing factors of the base plate thickness $t_{bp}$ relative to the wall plate thickness $t$, denoted as $i_t$, and the increasing factors of the base plate radius $R_{bp}$ relative to the pier section radius $R$,

denoted as $i_R$, are considered as design parameters to be investigated. The calculations for $i_t$ and $i_R$ are obtained using Equation (3) and Equation (4), respectively.

$$i_t = \frac{t_{bp} - t}{t} \tag{3}$$

$$i_R = \frac{R_{bp} - R}{R} \tag{4}$$

Furthermore, this paper introduces the prestressing ratio $\alpha$ and the self-recovery index $\lambda$ [26] to quantitatively study the effects of prestressed tendons and energy dissipation components on the seismic performance of the self-centering rocking steel bridge pier. The prestressing ratio $\alpha$ and the self-recovery index $\lambda$ are defined using Equations (5) and (6), respectively.

$$\alpha = \frac{\sigma_{p0}}{f_y} \tag{5}$$

$$\lambda = \frac{F_{sy}}{F_{cy}} \tag{6}$$

In the above equations, $\sigma_{p0}$ represents the normal stress induced by the initial prestress of the tendons, and $f_y$ represents the yield stress of the steel material. $F_{sy}$ and $F_{cy}$ represent the horizontal yielding forces of the bridge pier without energy dissipation and the energy dissipation elements, respectively. Their contributions to the total lateral resistance of the rocking pier are shown in Figure 2. In this study, external buckling-restrained energy dissipation steel plates were placed on the outside of the base plates, and $F_{cy}$ can be calculated using Equation (7):

$$F_{cy} = \sigma_{cy} A_c \frac{2R_{bp}}{h} \tag{7}$$

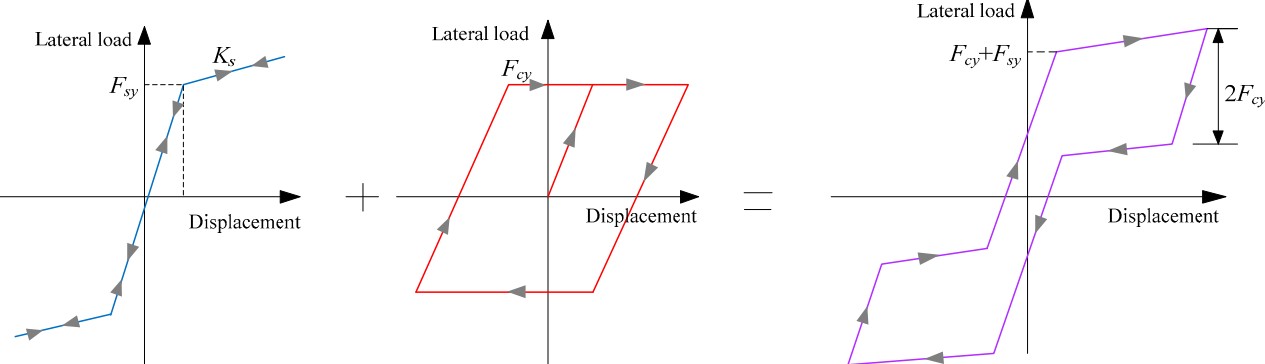

**Figure 2.** Schematic diagram of hysteresis curve of rocking steel bridge pier.

In the above equation, $\sigma_{cy}$ represents the yield strength of the buckling-restrained energy dissipation steel plate, and $A_c$ represents the cross-sectional area of a single energy dissipation steel plate. $F_{sy}$ can be approximately calculated based on the analytical method using the modified monolithic beam analogy [6], but the details are not discussed in this paper. To ensure that there is no excessive residual displacement, the recovery point B should be located above the displacement coordinate axis; hence, $\lambda$ should be greater than 1.0 [22].

After repeated rocking motions, the bottom part of the reinforced concrete rocking pier undergoes inevitable local crushing damage, leading to a decrease in lateral stiffness. Therefore, many studies have adopted structural measures such as external steel jackets to prevent excessive local pressure on the concrete. On the other hand, the steel bridge pier is a thin-walled structure, and during repeated rocking motions, the height of the compressed zone varies, which can result in elliptical deformation at the base plates and local buckling

deformation at the bottom of the wall plates [24]. To avoid excessive deformation and damage to the pier, this paper proposes reinforcement measures in the bottom area of the pier. The reinforcement methods include increasing the thickness of the wall plate and adding additional longitudinal stiffeners. In this paper, the additional stiffeners are placed on the inner side of the wall plate at every 45-degree angle. To measure the strength and influence of the additional longitudinal stiffeners on the section, an equivalent section without the stiffeners is introduced, which maintains the same section shape and the plastic moment $M_p$ [27], as shown in Figure 3. Based on the reinforcement method adopted in this paper, the equivalent thickness of the section without the stiffeners $\bar{t}$, corresponding to the pier with additional longitudinal stiffeners, can be obtained by equating it to the plastic moment of the pier with thickened wall plates, i.e.,

$$M_p = \left[4R^2t + \left(1 + \sqrt{2}\right)\left(2Rl - l^2\right)t_s\right]\sigma_y = 4R^2\bar{t}\sigma_y \tag{8}$$

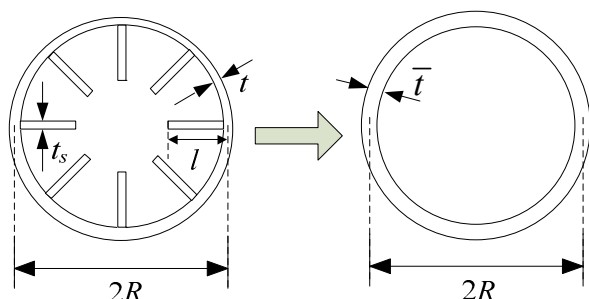

**Figure 3.** Equivalent wall thickness of piers with additional longitudinal stiffeners.

In the above equation, $t_s$ represents the thickness of the additional longitudinal stiffeners, and $l$ represents the width of the additional longitudinal stiffeners. In this paper, the radius-to-thickness ratio of the wall plate at the bottom of the plastic region or the radius-to-thickness ratio corresponding to the equivalent wall plate thickness $\bar{t}$ for the bridge piers with additional longitudinal stiffening ribs $R_{td}$ and the height of the thickened zone $l_d$ are taken as parameters of the study.

Based on the design parameters that influence the seismic performance of the rocking steel bridge piers mentioned above, this study focuses on the rocking steel bridge piers with different structural parameters, as presented in Table 1. Numerical simulations are conducted to analyze the quasi-static behavior of the piers under horizontal cyclic loads.

**Table 1.** Calculation components of rocking steel bridge piers.

| No. | $R \times t \times h$ (mm) | $R_t$ | $\bar{\lambda}$ | $N/N_y$ | $i_t$ (%) | $i_R$ (%) | $\alpha$ | $\lambda$ | Reinforcement Method | $R_{td}$ | $l_d$ (m) |
|---|---|---|---|---|---|---|---|---|---|---|---|
| 1 | $305 \times 10 \times 2100$ | 0.096 | 0.2711 | 0.10 | 0 | 10 | 0.10 | 1.763 | / | 0.096 | / |
| 2-A | $305 \times 10 \times 2100$ | 0.096 | 0.2711 | 0.10 | 100 | 0 | 0.10 | 1.763 | / | 0.096 | / |
| 2 | $305 \times 10 \times 2100$ | 0.096 | 0.2711 | 0.10 | 100 | 10 | 0.10 | 1.763 | / | 0.096 | / |
| 2-B | $305 \times 10 \times 2100$ | 0.096 | 0.2711 | 0.10 | 100 | 20 | 0.10 | 1.763 | / | 0.096 | / |
| 3 | $305 \times 10 \times 2100$ | 0.096 | 0.2711 | 0.10 | 200 | 10 | 0.10 | 1.763 | / | 0.096 | / |
| 4 | $305 \times 10 \times 2100$ | 0.096 | 0.2711 | 0.10 | 300 | 10 | 0.10 | 1.763 | / | 0.096 | / |
| 5 | $305 \times 10 \times 2100$ | 0.096 | 0.2711 | 0.10 | 100 | 10 | 0.10 | 1.763 | Thickened wall plate | 0.048 | 0.15 |
| 6 | $305 \times 10 \times 2100$ | 0.096 | 0.2711 | 0.10 | 100 | 10 | 0.10 | 1.763 | Thickened wall plate | 0.048 | 0.30 |
| 7 | $305 \times 10 \times 2100$ | 0.096 | 0.2711 | 0.10 | 100 | 10 | 0.10 | 1.763 | Thickened wall plate | 0.063 | 0.30 |
| 8 | $305 \times 10 \times 2100$ | 0.096 | 0.2711 | 0.10 | 100 | 10 | 0.10 | 1.763 | Additional stiffeners | 0.048 | 0.15 |
| 9 | $305 \times 10 \times 2100$ | 0.096 | 0.2711 | 0.10 | 100 | 10 | 0.10 | 1.763 | Additional stiffeners | 0.048 | 0.30 |
| 10 | $305 \times 10 \times 2100$ | 0.096 | 0.2711 | 0.10 | 100 | 10 | 0.10 | 1.763 | Additional stiffeners | 0.063 | 0.30 |
| 11 | $305 \times 10 \times 2100$ | 0.096 | 0.2711 | 0.20 | 100 | 10 | 0.10 | 1.763 | / | 0.096 | / |
| 12 | $305 \times 10 \times 5100$ | 0.096 | 0.6584 | 0.10 | 100 | 10 | 0.10 | 1.763 | / | 0.096 | / |
| 13 | $305 \times 20 \times 2100$ | 0.048 | 0.2711 | 0.10 | 100 | 10 | 0.10 | 1.763 | / | 0.048 | / |
| 14 | $305 \times 10 \times 2100$ | 0.096 | 0.2711 | 0.10 | 100 | 10 | 0.20 | 1.763 | / | 0.096 | / |
| 15 | $305 \times 10 \times 2100$ | 0.096 | 0.2711 | 0.10 | 100 | 10 | 0.10 | 0.882 | / | 0.096 | / |

### 2.2. Finite Element Analysis Model and Verification

To systematically investigate the seismic performance of each rocking steel bridge pier as shown in Table 1, a refined finite element model was established using the commercial software Abaqus 6.14. Figure 4 illustrates the finite element model of the pier.

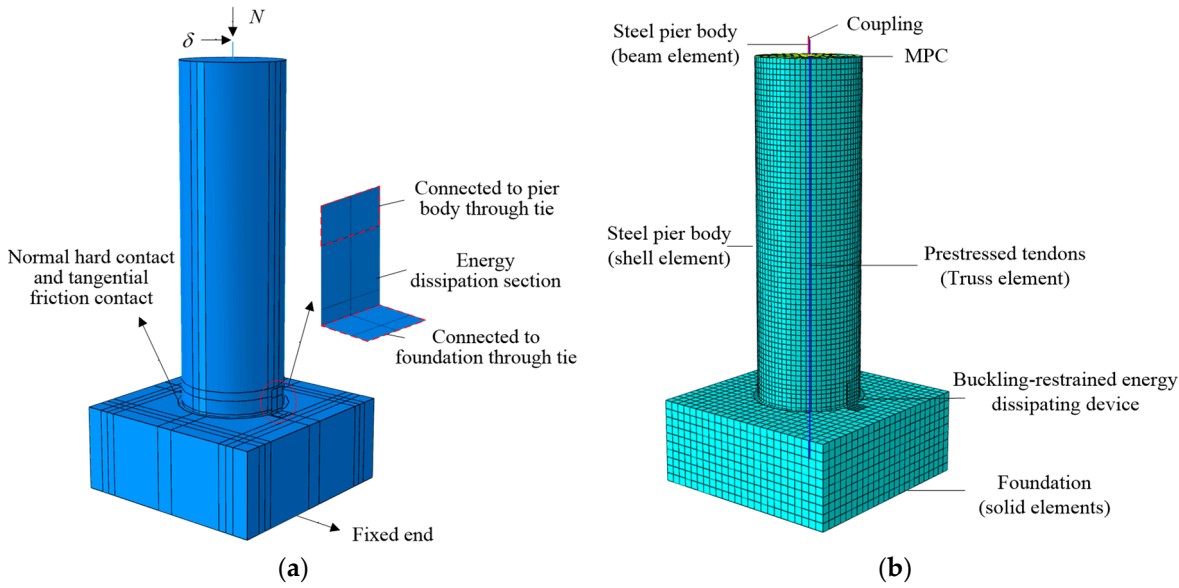

**Figure 4.** Finite element model of circular-section rocking steel bridge pier: (**a**) Geometric model (**b**) Element discrete method.

In the model, a rigid short beam is used to simulate the top portion of the pier. A horizontal cyclic displacement $\delta$ and a constant axial force $N$ was applied at the top of this short beam. The bottom of the short beam is connected to the shell elements below using multi-point constraints (MPCs). The interaction along the normal direction between the pier base and the concrete foundation is modeled through hard contact, while the interaction along the tangential direction is modeled with frictional contact using a friction coefficient of 0.2 [28]. One end of the prestressed tendons is connected to the foundation surface using MPC, while the other end is coupled to the upper end point of the short beam (i.e., the top of the pier) using a two-point coupling connection. The L-shaped buckling-restrained energy dissipating device is connected to one side of the pier using the tie connection, and the other side is connected to the foundation surface using the tie connection, while restraining its out-of-plane deformation. Meanwhile, the bottom of the foundation is fixed.

The loading process consists of three static analysis steps. Step 1 applies the constant axial force $N$ and the self-weight of the pier, step 2 applies the initial prestressing forces through the temperature reduction method, and step 3 applies the horizontal cyclic displacement $\delta$.

In the FE model, Q345qC steel is used for the pier, and its hysteretic constitutive model adopts the Chaboche cyclic hardening hysteresis model that considers the Bauschinger effect and the cyclic reinforcement effect. Material properties and the parameters of the Chaboche cyclic hardening hysteresis model for Q345qC steel have been calibrated in reference [29] and are directly utilized in this study. Q235 steel is employed for the energy dissipating component. The yield strength of Q235 steel is 235 MPa and the ideal elastoplastic model was employed as the hysteretic constitutive model of Q235 steel. Both the steel bridge pier and the energy dissipating component are simulated using four-node reduced integration shell elements (S4R). The prestressed tendons are simulated using truss elements (T3D2), while the reinforced concrete foundation is simulated using eight-node reduced integration solid elements (C3D8R).

This paper adopts displacement control to perform horizontal cyclic quasi-static loading on the bridge pier. Figure 5 shows the displacement loading system, where the *Y*-axis represents the normalized displacement $\delta/\delta_y$, and the *X*-axis represents the number of cycles *n*. The peak displacement value of each cycle increases incrementally, with the displacement increment being the yield displacement of the pier.

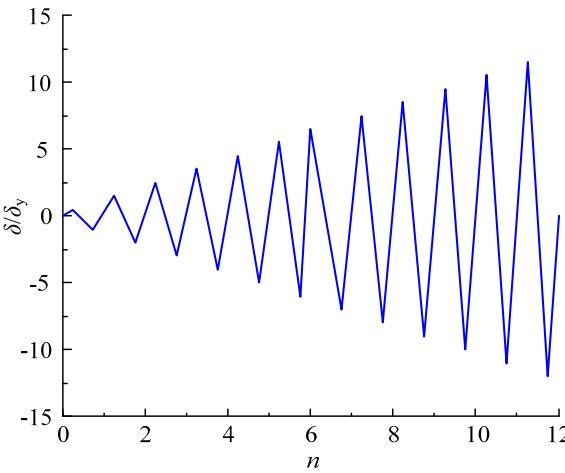

**Figure 5.** Quasi-static loading system.

To validate the effectiveness of the finite element analysis model used in this study, numerical simulations were conducted on the test rocking steel bridge pier with a base plate, designated as RP3-DT43-PT44-BP20 in reference [24]. The basic parameters of the pier are listed in the Table 2. The test pier was an unreinforced pier at the bottom, and the calibrated results of the Chaboche hysteresis model parameters for the steel material used are provided in reference [25], which were directly incorporated into the FE model in this study.

**Table 2.** Design parameters of experimental rocking steel bridge pier [24].

| Specimen No. | $R \times t \times h$ (mm) | $R_t$ | $\bar{\lambda}$ | $\alpha$ | $\lambda$ | $N/N_y$ | $i_t$ (%) (%) | $i_R$ (%) (%) |
|---|---|---|---|---|---|---|---|---|
| RP3-DT43-PT44-BP20 | $203 \times 9.53 \times 1677$ | 0.071 | 0.335 | 0.162 | 0.0 | 0.0 | 166.5 | 25.1 |

To analyze the influence of pier element size on the calculation results, FE models of the test rocking steel bridge pier were established with shell element sizes of 15 mm, 40 mm, and 70 mm, respectively, for grid sensitivity analysis. The hysteresis curves are compared in Figure 6. From the computed results in the figure, it can be observed that the shell element size has minimal impact on the calculation results. Therefore, for subsequent studies in this paper, a uniform shell element size of 40 mm will be adopted.

The comparison between the hysteresis curves obtained from the FE model and the test results as well as the stress-displacement calculation results of the prestressed tendons are shown in Figure 7. The results indicate that due to the absence of energy dissipation devices in this pier, the area enclosed by the hysteresis curve is small, and there is a slight plastic energy dissipation at the base of the pier. Since the prestress loss caused by prestressing relaxation at the anchorage end [24,25] is not considered in the computational model used in this study, there is no stress decrement at the displacement zero point. In the figure, $f_{PT}$ is the stress of the prestressed tendon and $f_{PT,u}$ is the ultimate tensile stress of the pre-stressed tendon. From the computational results, it can be observed that the load-displacement curve and the stress-displacement curve are very close to the experimental results. Therefore, adopting the FE model proposed in this study can achieve accurate simulation results of the seismic performance of rocking steel bridge piers. The same FE

modeling approach will be employed in the subsequent study on the seismic performance of the piers.

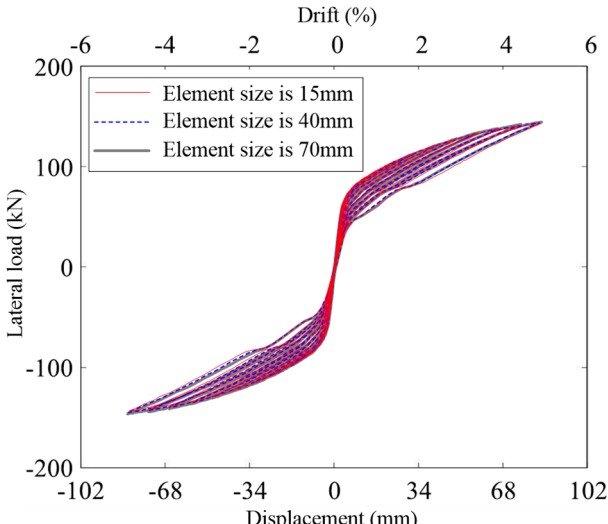

**Figure 6.** Influence of shell element size on the hysteresis curve of the rocking steel bridge pier.

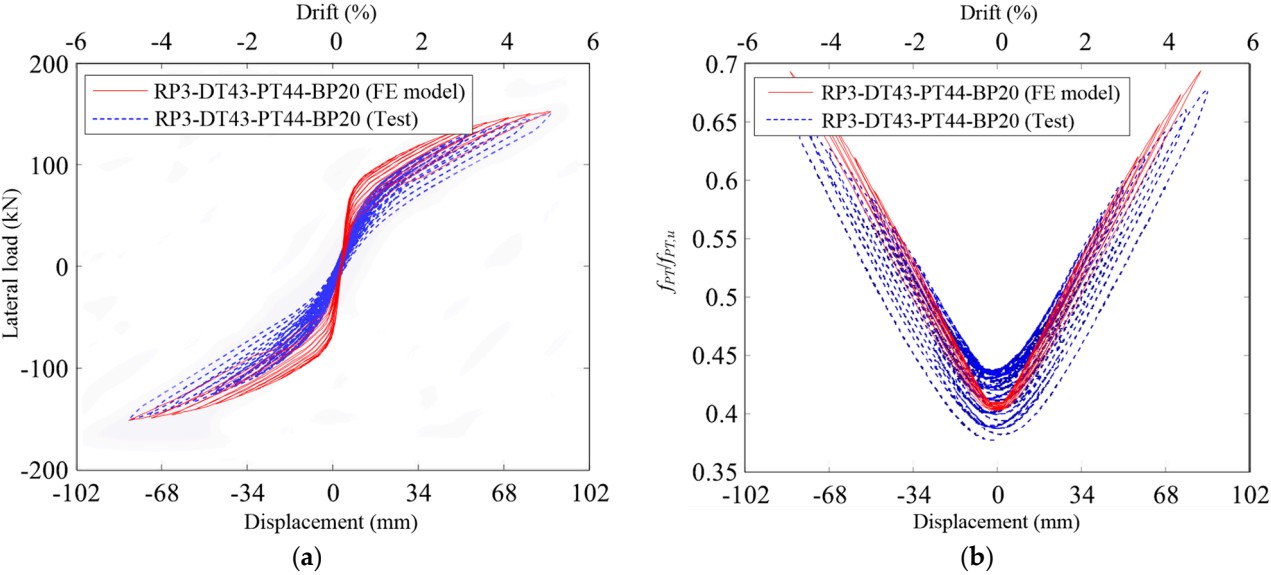

**Figure 7.** Comparison of FE analysis results and experimental results: (**a**) Comparison of load-displacement hysteresis curves (**b**) Comparison of prestressed tendon stress-displacement results.

## 3. Comparison of Seismic Performance between Concrete-Filled Rocking Steel Bridge Piers and Hollow Rocking Steel Bridge Piers

The concrete-filled rocking steel bridge pier have received significant attention from experts and scholars [13,16]. This paper takes the Pier 2 and Pier 12 in Table 1 as examples to carry out a comparative analysis of the seismic performance of the concrete-filled and the hollow rocking steel bridge piers. The piers No. 2 and No. 12 only differ in terms of their slenderness ratios, representing relatively low and high piers, respectively. C30 concrete, commonly used in China, is selected to fill the steel bridge pier, forming concrete-filled rocking steel bridge pier. The axial compressive strength of C30 concrete is 20.1 MPa and the elastic modulus is 22,500 MPa. The elastoplastic constitutive relationship of the concrete adopts the concrete plastic damaged model. The interface between the inner filled concrete and the outer steel tube is modeled using hard contact and tangential friction, with a

friction coefficient of 0.2 [28]. The computational models take into account the self weight of the piers.

Figure 8 presents a comparison of the hysteresis curves between concrete-filled rocking steel bridge piers and hollow rocking steel bridge piers. The results indicate that under horizontal cyclic loading, both rocking steel bridge piers without filled concrete and concrete-filled rocking steel bridge pier exhibit typical flag-shaped hysteresis curves with small residual displacements. This signifies that the piers possess good energy dissipation capacity and self-centering performance. Furthermore, whether it is a low-height or high-height rocking pier, the hysteresis curve results with and without concrete filling are very similar. This is because during intense earthquakes, the piers primarily rely on external steel plates for energy dissipation, and the degree of plastic deformation in the pier body is relatively small, resulting in minimal overall differences in the hysteresis curves. Hence, employing hollow steel bridge piers as pier bodies can effectively achieve energy dissipation and self-centering functionality in rock piers, while enhancing the cost-effectiveness of the structural design.

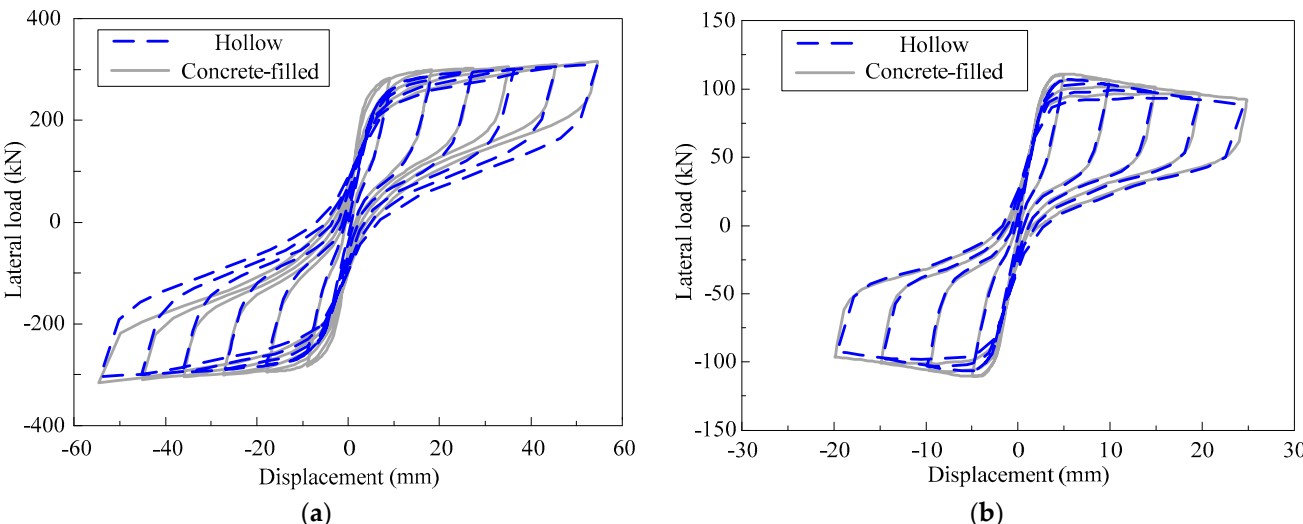

**Figure 8.** Comparison of hysteresis curve of hollow rocking steel bridge piers and concrete-filled rocking steel bridge piers: (**a**) Pier 2; (**b**) Pier 12.

## 4. The Influence of Overall Design Parameters of Bridge Piers on the Seismic Performance of Rocking Steel Bridge Piers

This article adopts the modeling method described in Section 1 to conduct seismic performance analysis on each pier listed in Table 1. In this section, the influence of various overall design parameters on the seismic performance will be analyzed separately.

The energy dissipation *E* during each cyclic loading of the bridge pier is calculated using the following equation.

$$E = \int_{\delta_{\min}}^{\delta_{\max}} \left( F^+(\delta) - F^-(\delta) \right) \mathrm{d}\delta \tag{9}$$

In the above equation, $\delta_{\min}$ and $\delta_{\max}$ denote the initial and final displacements, respectively, for the current cycle. $F^+(\delta)$ and $F^-(\delta)$ represent the loading restoring force and unloading restoring force, respectively, at the same displacement of $\delta$.

### 4.1. Self-Centering Index

Figure 9 shows a comparison of the hysteresis curves between Pier 2 and Pier 15. It can be observed that due to a self-recovery indicator $\lambda$ less than 1.0 for Pier 15, the overall energy dissipation capacity is significantly enhanced compared to Pier 2, increasing by 85.1%. However, the reverse unloading point will be located below the displacement

coordinate axis, leading to a significant increase in residual displacement, which increases by 170%. This indicates that appropriate dimensions of energy dissipation elements should be selected in the design to achieve a balance between energy dissipation capacity and residual displacement.

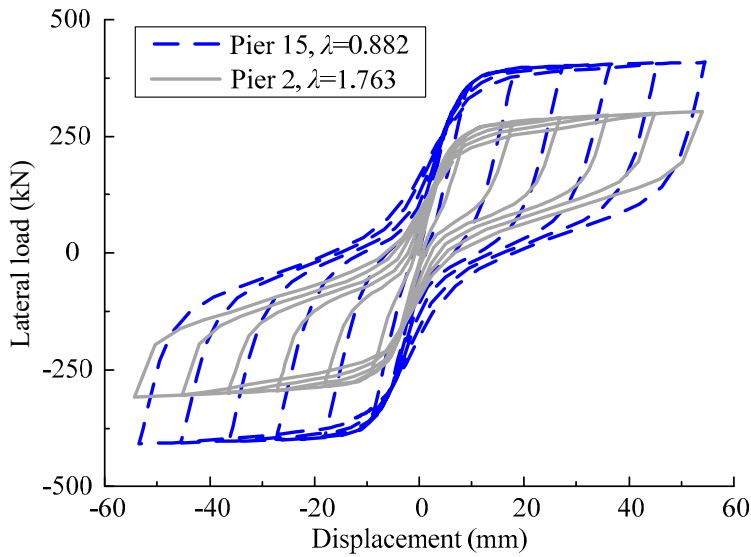

**Figure 9.** Impact of self-centering index.

### 4.2. Prestressing Ratio

Figure 10 presents a comparison of the hysteresis curves between Pier 2 and Pier 14. When the prestressing ratio decreases from 1.763 to 0.882, it can be observed that the horizontal bearing capacity of the pier is significantly enhanced, increasing by 20%. However, there is also an increase in residual displacement of the pier.

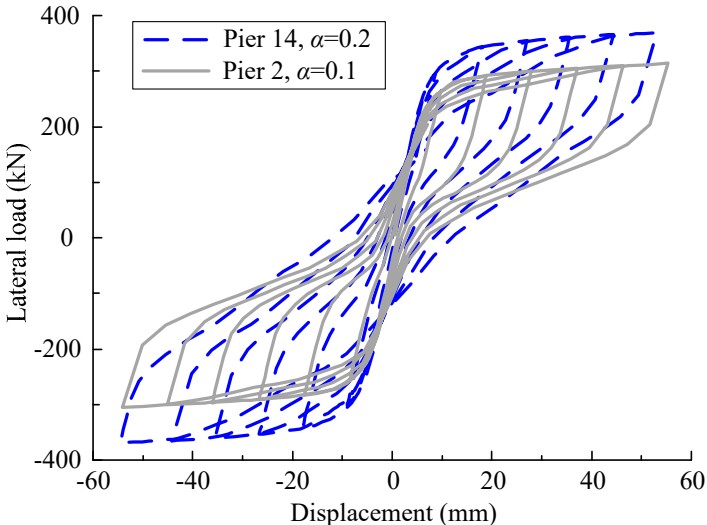

**Figure 10.** Impact of prestressing ratio.

### 4.3. Base Plate Thickness Increase Factor

Figure 11 illustrates a comparison of the hysteresis curves and local deformation of the wall plate at the bottom of the pier at the maximum displacement moment for Piers 1–4 in Table 1. For Pier 1, with 0% $i_t$, there is a significant local instability deformation of the wall plate at the bottom of the pier, leading to a rapid decrease in bearing capacity. When $i_t$ increases to 100%, the degree of local deformation significantly decreases, with an increase of 6.7% in energy dissipation capacity and a decrease of 9.1% in residual displacement.

However, as $i_t$ continues to increase to 300%, there is not much change in the extent of local deformation of the wall plate at the bottom of the pier, with a reduction of only 1.5% in energy dissipation compared to the condition in Pier 2 and a decrease of 1.4% in residual displacement. The reinforcement effect is not significant.

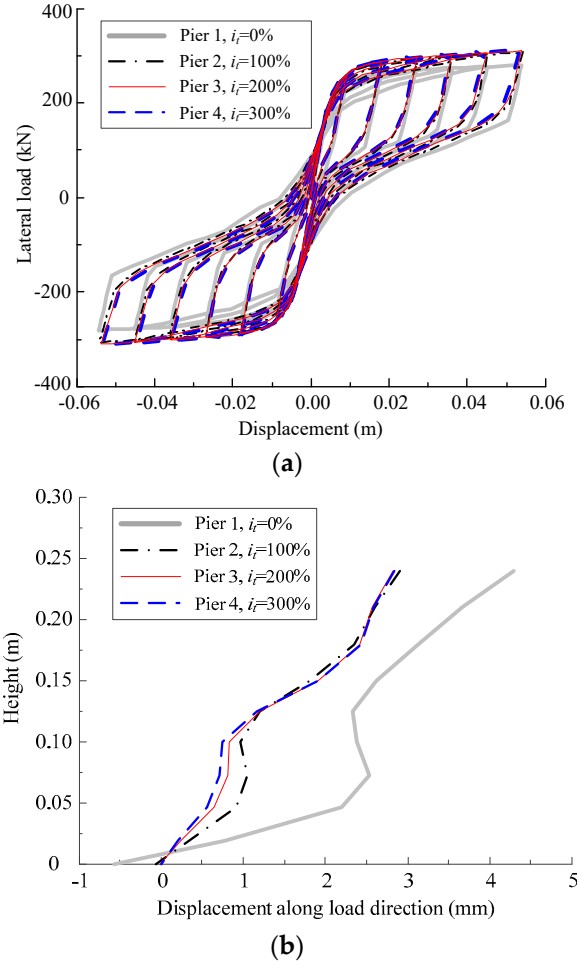

**Figure 11.** Influence of base plate thickness increase factor: (**a**) Hysteresis curve comparison; (**b**) Local deformation wall plate at the bottom of the pier.

Therefore, in the design of rocking steel bridge piers, the thickness of the base plate should be significantly greater than that of the wall plate, but not excessively large. In this study, the coefficient of increasing the base plate thickness, $i_t$, is kept at 100% for the analysis of other piers.

### 4.4. Base Plate Radius Increase Factor

Figure 12 shows a comparison of the hysteresis curves and vertical deformation of the base plates for Piers 2-A, 2, and 2-B. For Pier 2-A, the $i_R$ is 0%, which means that the base plate radius is the same as the pier radius, and the base plate undergoes an upward concave deformation, resulting in a lower overall bearing capacity of the pier. When $i_R$ is 10%, the base plate within the cross-sectional area of the pier does not undergo significant deformation, but the exposed portion of the base plate experiences an upward deflection, causing an elliptical deformation of the base plate. At this point, the bearing capacity of the pier increases by 4.9%. When $i_R$ continues to increase to 20%, there is no significant change in the deformation pattern of the base plate, and the bearing capacity remains unchanged.

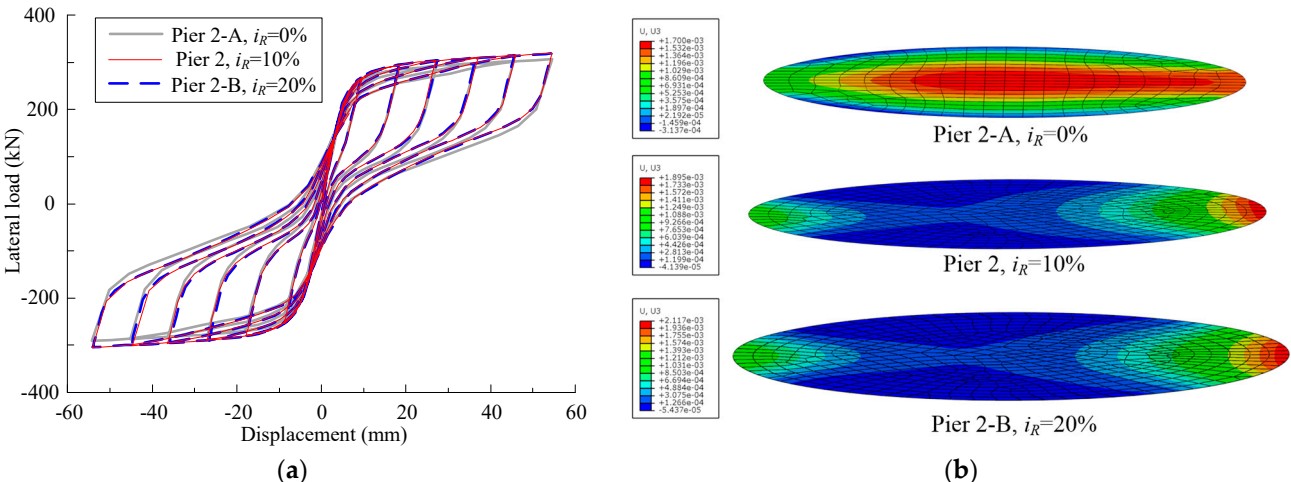

**Figure 12.** Influence of base plate radius increase factor: (**a**) Hysteresis curve comparison; (**b**) Vertical deformation comparison of base plate.

Therefore, in the design of rocking steel bridge piers, it is necessary to have the base plate radius slightly larger than the pier radius. In this study, the coefficient of increasing the base plate radius, $i_R$, is kept at 10% for the analysis of other piers.

### 4.5. Axial Compression Ratio

Figure 13 compares the load-displacement hysteresis curves for Piers 2 and 11. Due to a higher axial compression ratio, Pier 11 has a greater initial stiffness and peak bearing capacity than Pier 2. However, as the loading progresses, the second-order effects become significant, leading to faster degradation of the bearing capacity of Pier 11.

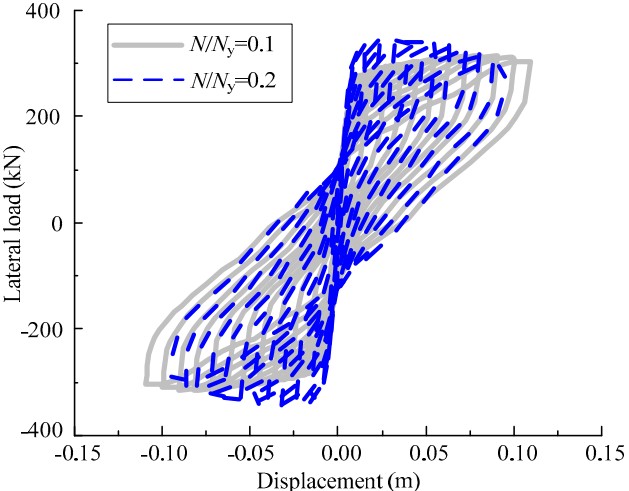

**Figure 13.** Impact of axial compression ratio.

### 4.6. Slenderness Ratio

Figure 14 shows a comparison of the hysteresis curves for Piers 2 and 12. Due to the increased height of Pier 12 compared to Pier 2, the horizontal bearing capacity of Pier 12 is significantly lower, by 141%, and it is more prone to overturning. The stiffness degradation effect is also evident. Therefore, the aspect ratio of rocking piers should be controlled within a reasonable range.

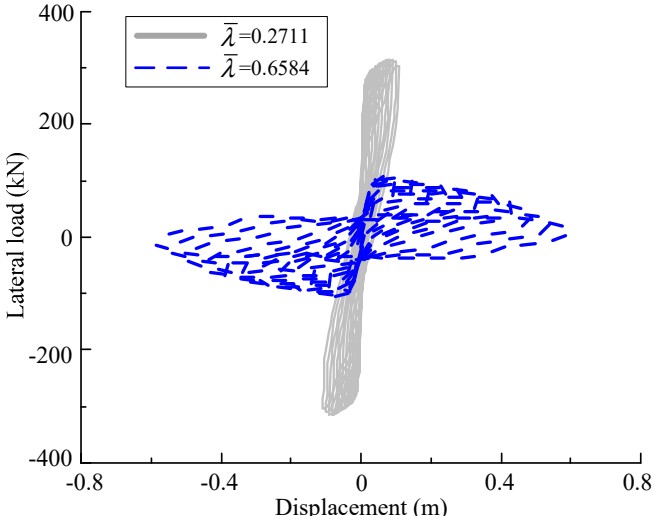

**Figure 14.** Impact of slenderness ratio.

### 4.7. Radius-to-Thickness Ratio

Figure 15 compares the hysteresis curves for Piers 2 and 13. To keep the prestressing ratio and axial compression ratio unchanged, Pier 13 also increases the sectional area of prestressing tendons and axial compression compared to Pier 2. With a wall thickness of 20 mm, Pier 13 dissipates the seismic energy through both the pier body and the energy-dissipating steel plates, resulting in a significantly higher bearing capacity, 76.5%, compared to Pier 2 with a wall thickness of 10 mm.

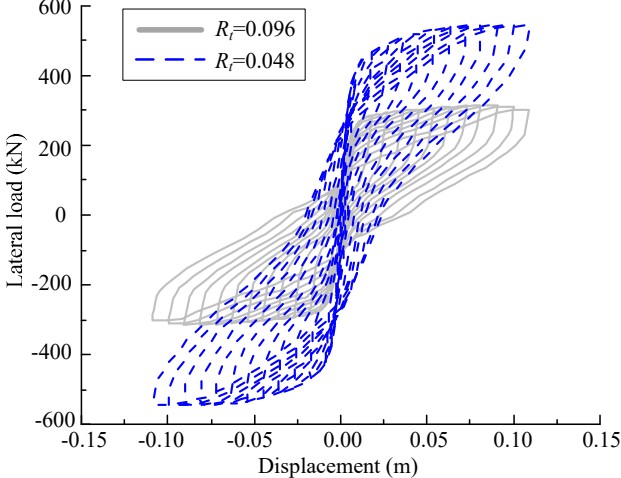

**Figure 15.** Impact of diameter-to-thickness ratio.

## 5. Seismic Strengthening Mechanism Analysis of the Plastic Region

To analyze the seismic strengthening mechanism of the plastic region at the bottom of the piers, a comparative analysis was conducted between the unreinforced Pier 2 and the reinforced Piers 6 and 7 with thickened wall plates, as well as Piers 9 and 10 with additional longitudinal stiffeners. The hysteresis curves are shown in Figure 16. It can be observed that both the reinforcement methods, i.e., thickened wall plates and additional stiffeners, resulted in a decrease in the thickness-to-radius ratio $R_{td}$ in the bottom region of the piers. This significantly enhanced the ductility performance of the piers, reduced the residual displacement, and mitigated the stiffness degradation effects.

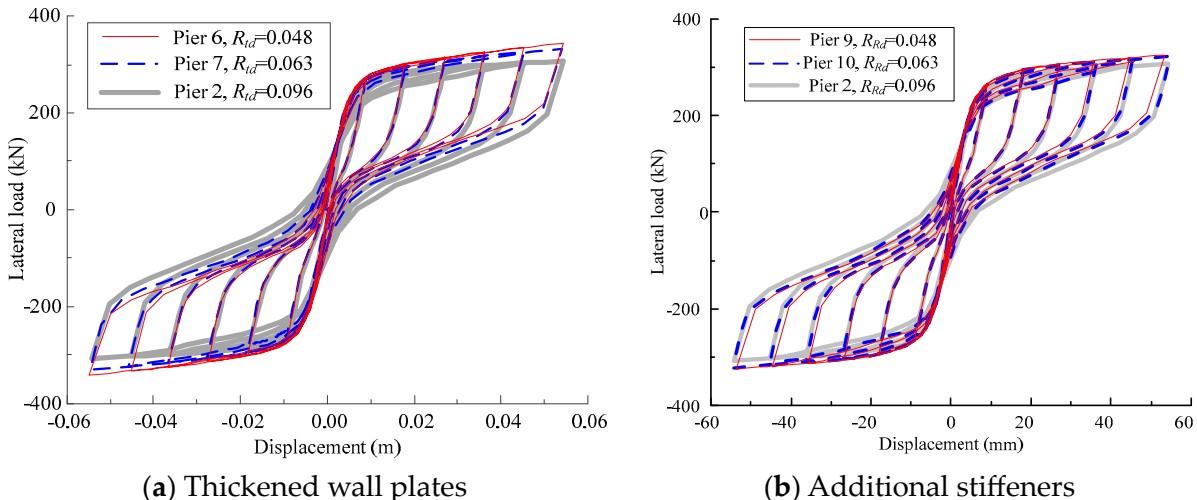

**(a)** Thickened wall plates

**(b)** Additional stiffeners

**Figure 16.** Impact of pier bottom diameter-to-thickness ratio.

To analyze the effect of reinforcement zone length on the hysteresis performance, a comparative analysis was conducted between the unreinforced Pier 2 and the reinforced Piers 5 and 6 with different heights of thickened wall plates, as well as reinforced Piers 8 and 9 with different heights of additional stiffeners. The hysteresis curves are presented in Figure 17. It can be seen that the reinforcement within a range of 0.3 m exhibited further improved performance compared to the reinforcement within a length of only 0.15 m.

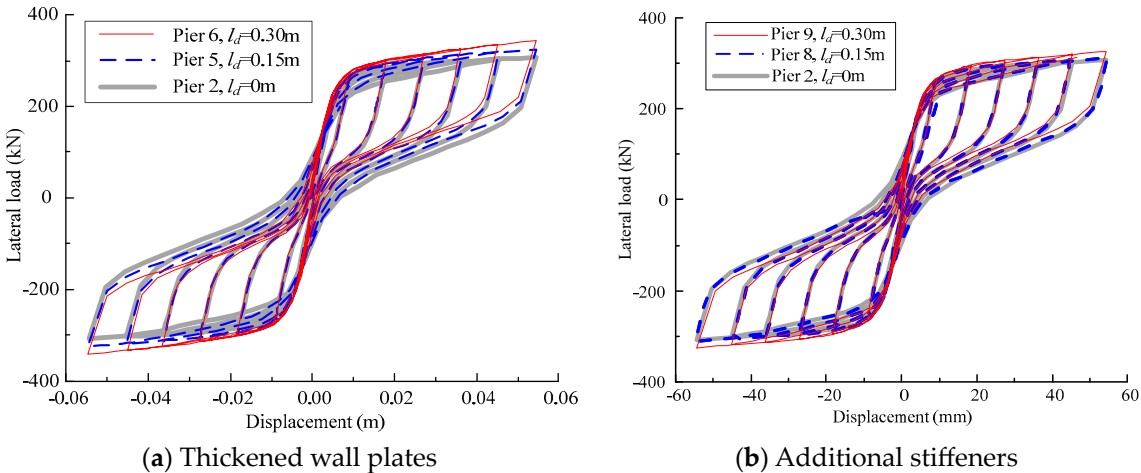

**(a)** Thickened wall plates

**(b)** Additional stiffeners

**Figure 17.** Impact of reinforcement zone length.

To further compare the effectiveness of the additional stiffeners and thickened wall plates at the bottom of the piers in enhancing the seismic performance of bridge piers, a comparison of the hysteresis curves was carried out between the unreinforced Pier 2 and the reinforced Piers 5 and 8 with thickened wall plates and additional stiffeners within a range of 0.15 m, as well as Piers 6 and 9 with thickened wall plates and additional stiffeners within a range of 0.3 m. This is illustrated in Figure 18. From the figure, it can be observed that under the same reinforcement length and radius-to-thickness ratio, the load-carrying capacity of the piers with thickened wall plates is significantly higher than that of the piers with additional longitudinal stiffeners. This indicates that the thickened wall plate approach is a more effective method for seismic strengthening. Figure 19 further presents the comparative results of the load-carrying capacity, energy dissipation capacity, initial lateral stiffness, and residual displacement under the condition of bottom reinforcement within a range of 0.15 m. Based on the comparison between Piers 2 and 5, it can be observed

that when thickened wall plates are applied, the maximum load-carrying capacity of the rock piers increases by 5.5%, energy dissipation value increases by 0.18%, initial stiffness increases by 8.9%, and residual displacement decreases by 15.6%. On the other hand, based on the comparison between Piers 2 and 8, when additional stiffeners are utilized, the maximum load-carrying capacity of the rock piers increases by 0.18%, energy dissipation value decreases by 1.4%, initial stiffness increases by 5.1%, and residual displacement decreases by 9.8%.

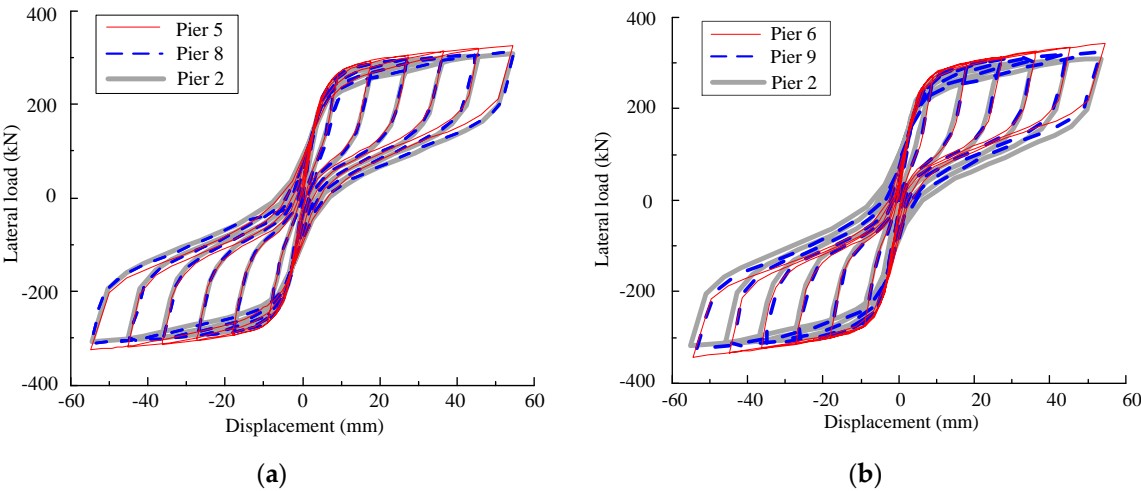

**Figure 18.** Comparison of hysteresis curves using different reinforcement methods. (**a**) Reinforcement within 0.15 m; (**b**) Reinforcement within 0.30 m.

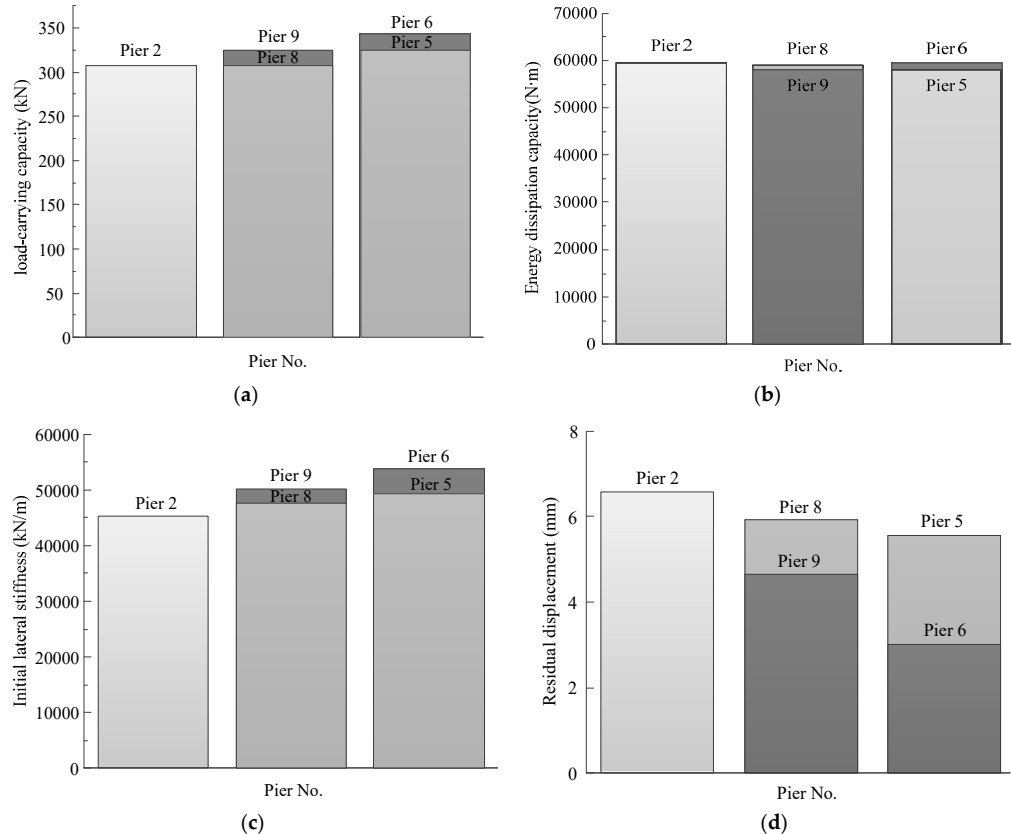

**Figure 19.** Comparison of calculation results using different reinforcement methods within a range of 0.15 m. (**a**) Load-carrying capacity; (**b**) Energy dissipation capacity; (**c**) Initial lateral stiffness; (**d**) Residual displacement.

Figure 19 also presents the comparative results of the load-carrying capacity, energy dissipation capacity, elastic lateral stiffness, and residual displacement of the piers under the condition of bottom reinforcement within a range of 0.3 m. Based on the comparison between Piers 2 and 6, when thickened wall plates are applied within a range of 0.3 m, the maximum load-carrying capacity of the piers increases by 11.5%, energy dissipation value decreases by 0.1%, initial stiffness increases by 18.9%, and residual displacement decreases by 54.7%. Similarly, based on the comparison between Piers 2 and 9, when additional stiffeners are utilized within a range of 0.3 m, the maximum load-carrying capacity of the rock piers increases by 5.8%, energy dissipation value decreases by 2.6%, initial stiffness increases by 11.1%, and residual displacement decreases by 31.4%.

From the comprehensive analysis, it can be concluded that regardless of the range, 0.15 m or 0.3 m, the reinforcement method of thickened wall plates is superior to additional stiffeners in improving the seismic performance. Further analysis will be conducted to explore the mechanism behind the enhancement in seismic performance via the thickened bottom wall plate reinforcement.

Figure 20 compares the elastic lateral stiffness trends, residual displacement trends, and energy dissipation capacity of Piers 5, 6, and 7, which were reinforced with thickened wall plates, and the unreinforced Pier 2. Overall, the effect of thickened wall plates on the cyclic energy dissipation of the self-centering rocking steel bridge piers is not significant. However, after thickening the wall plates, the growth of residual displacement becomes slower and the lateral stiffness increases for each cycle, significantly improving the seismic capacity of the piers.

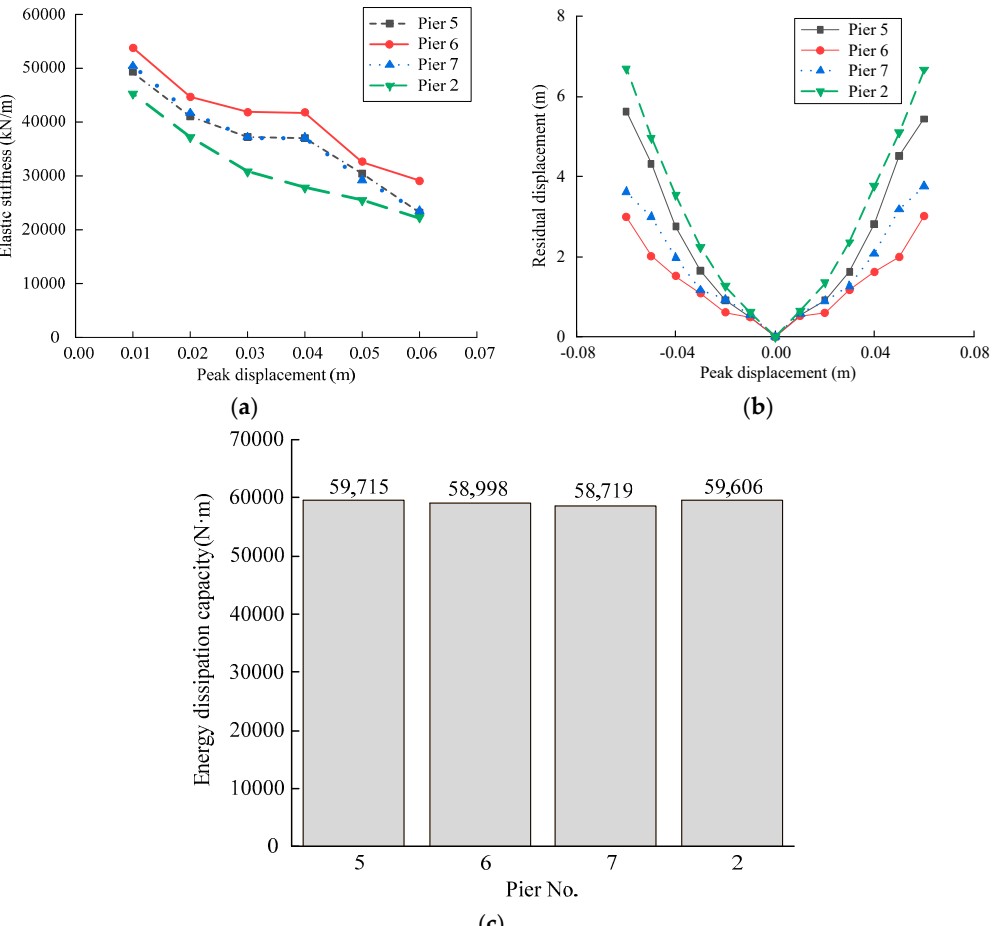

**Figure 20.** Comparison of performance indicators of piers with thickened wall plates: (**a**) Elastic stiffness trends; (**b**) Residual displacement trends; (**c**) Energy dissipation capacity.

Based on the comparison between Specimens 2 and 5, when the radius-to-thickness ratio of the wall plates within a range of 0.15 m decreases from 0.096 to the 0.048, the maximum load-carrying capacity of the rock piers increases by 5.5%, the maximum residual displacement decreases by 18.2%, and the initial stiffness increases by 8.9%. According to the comparison between Specimens 5 and 6, when the reinforced area with thickened wall plates is further increased to 0.3 m, the maximum load-carrying capacity increases by 5.7%, the residual displacement decreases by 45%, and the initial stiffness increases by 9.2%.

On the other hand, based on the comparison between Specimens 2 and 7, when the radius-to-thickness ratio of the wall plates within a 0.3 m reinforcement range decreases to 0.063, the maximum load-carrying capacity of the piers increases by 7.7%, the maximum residual displacement decreases by 43.6%, and the initial stiffness increases by 11.4%. According to the comparison between Piers 6 and 7, when the diameter-to-thickness ratio of the wall plates decreases to 0.048, the maximum load-carrying capacity increases by 3.5%, the residual displacement decreases by 20%, and the initial stiffness increases by 6.7%.

Therefore, when reinforcing self-centering steel bridge piers for seismic performance, attention should be paid to both the thickness of the reinforced steel plates and the height of the reinforcement. A larger reinforcement height is more effective in reducing the residual displacement of the piers compared to thickening the steel plates alone.

## 6. Conclusions

To study the seismic performance of rocking steel bridge piers, a high-precision FE computational model suitable for such structures was established for numerical simulation analysis. Based on the calculation results and theoretical analysis, the following conclusions can be drawn:

(1) A comparison with the existing pseudo-static test results shows that the FE model used in this study achieves a high level of accuracy in simulating the horizontal load-displacement curve of rocking steel bridge piers;

(2) Since the pier body is only used for energy dissipation peripherally and exhibits low plastic deformation, the hysteresis curves of concrete-filled rocking steel bridge piers and hollow rocking steel bridge piers are quite similar. Directly using hollow steel pier bodies improves the economic efficiency of the design;

(3) Increasing the self-recovery index $\lambda$ significantly enhances the energy dissipation capacity of the bridge pier, but care should be taken to prevent excessive residual displacement. With an increase in prestress ratio, both the bearing capacity and residual displacement of the pier increase. Significantly thicker base plate compared to the wall plate can significantly improve the pier's resistance to local instability. Making the radius of the base plate slightly larger than the cross-sectional radius of the pier avoids concave deformation of the base plate and increases the pier's bearing capacity by approximately 5%. An increase in axial compression ratio increases the additional moment, leading to early degradation of the bearing capacity;

(4) Thickening base plates and adding longitudinal stiffeners can enhance the seismic performance of rocking steel bridge piers. Regardless of reinforcement within the range of 0.15 m or 0.3 m, reinforcing the thickness of the wall plate yields better results compared to additional longitudinal stiffeners.

(5) When the wall plate is thickened as the diameter-to-thickness ratio reaches 0.048 within a range of 0.3 m, the maximum load-carrying capacity of the rock piers increases by 11.5%, energy dissipation decreases by 0.1%, initial stiffness increases by 18.9%, and the residual displacement decreases by 54.7%. Reinforcement at the bottom of the pier has little influence on energy dissipation capacity but can slow down the growth of residual displacement, increase the lateral stiffness of the pier, and have a positive effect on the seismic performance of the bridge. A greater reinforcement height is more effective in reducing residual displacement compared to thickening the wall plate.

**Author Contributions:** Conceptualization, H.Z.; funding acquisition, H.Z.; investigation, H.Z. and C.N.; validation, C.N. and R.D.; writing—original draft preparation, H.Z. and C.N.; writing—review & editing, H.Z.; project administration, Z.T.; funding acquisition, H.Z. All authors have read and agreed to the published version of the manuscript.

**Funding:** This research was funded by the General Program of National Natural Science Foundation of China, Grant No. 51878606 and China Postdoctoral Science Foundation, Grant No. 2023M730154.

**Institutional Review Board Statement:** Not applicable.

**Informed Consent Statement:** Not applicable.

**Data Availability Statement:** Not applicable.

**Conflicts of Interest:** The authors declare no conflict of interest.

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
