# Peer review of "Research on Seismic Performance and Reinforcement Methods for Self-Centering Rocking Steel Bridge Piers"

_applsci, doi:10.3390/app13169108_

Round 1
Reviewer 1 Report
Detailed comments for the authors can be found inside the attached PDF file.

Minor improvements to English language are needed.
Reviewer 2 Report
(1) Title: please change the “menthod” to “method”.
(2) Page 8, Lines #250 to #261: the reviewer does not agree with the author’s statement that the FE results are close to the experimental ones. Figure 7 shows the major discrepancies between FE and test data. More specifically, the stiffness in the hysteretic curves has significant differences. Similarly, the entire prestressed tendon stress-displacements drastically deviated from the test to FE simulation.
(3) Page 9, Lines #268 to #290: the author has verified the FE models with pure steel columns (without filling by concrete). However, the author subsequently analyzes a steel column with concrete. How does the author ensure that the FE model with concrete is reliable without verifying it against experimental data? For example, the FE model with concrete requires defining the interaction between steel tube and concrete, which is not validated.
(4) The author is suggested to improve the parametric study section further. For example, some sections (e.g., Sections 3.4 and 3.5) present “trivial” analysis results (i.e., results are self-evident without the need for adopting FE simulations)
Round 2
Reviewer 2 Report
The reviewer is delighted to see that the authors have addressed all comments properly.